# Comparative analysis of transferrin and IgG N-glycosylation in two human populations

Irena Trbojević-Akmačić [1,11], Frano Vučković [1,11], Tea Pribić[1], Marija Vilaj[1], Urh Černigoj[2], Jana Vidič[2], Jelena Šimunović[1], Agnieszka Kępka[1,3], Ivana Kolčić[4,5], Lucija Klarić [6], Mislav Novokmet[1], Maja Pučić-Baković [1], Erdmann Rapp [7,8], Aleš Štrancar[2], Ozren Polašek[4,5], James F. Wilson [6,9] & Gordan Lauc [1,10 ✉]

Human plasma transferrin (Tf) N-glycosylation has been mostly studied as a marker for congenital disorders of glycosylation, alcohol abuse, and hepatocellular carcinoma. However, inter-individual variability of Tf N-glycosylation is not known, mainly due to technical limitations of Tf isolation in large-scale studies. Here, we present a highly specific robust high-throughput approach for Tf purification from human blood plasma and detailed characterization of Tf N-glycosylation on the level of released glycans by ultra-high-performance liquid chromatography based on hydrophilic interactions and fluorescence detection (HILIC-UHPLC-FLD), exoglycosidase sequencing, and matrix-assisted laser desorption/ionization time-of-flight mass spectrometry (MALDI-TOF-MS). We perform a large-scale comparative study of Tf and immunoglobulin G (IgG) N-glycosylation analysis in two human populations and demonstrate that Tf N-glycosylation is associated with age and sex, along with multiple biochemical and physiological traits. Observed association patterns differ compared to the IgG N-glycome corroborating tissue-specific N-glycosylation and specific N-glycans' role in their distinct physiological functions.

[1] Genos Glycoscience Research Laboratory, Zagreb, Croatia. [2] BIA Separations d.o.o., a Sartorius company, Ajdovščina, Slovenia. [3] Department of Immunology, Faculty of Biology, Institute of Zoology, University of Warsaw, Warsaw, Poland. [4] Department of Public Health, University of Split School of Medicine, Split, Croatia. [5] Algebra University College, Zagreb, Croatia. [6] MRC Human Genetics Unit, Institute for Genetics and Cancer, University of Edinburgh, Edinburgh, UK. [7] Max Planck Institute for Dynamics of Complex Technical Systems, Magdeburg, Germany. [8] glyXera GmbH, Magdeburg, Germany. [9] Centre for Global Health Research, Usher Institute, University of Edinburgh, Edinburgh, UK. [10] Faculty of Pharmacy and Biochemistry, University of Zagreb, Zagreb, Croatia. [11] These authors contributed equally: Irena Trbojević-Akmačić, Frano Vučković. ✉email: glauc@pharma.hr

Transferrin (Tf) is an iron-binding glycoprotein produced mainly by hepatocytes whose key function is the regulation of free iron levels in biological fluids. In humans, Tf has two N-glycosylation sites—at asparagine 432 and asparagine 630. Glycans determine glycoprotein stability and function and are involved in protein and cell recognition and interaction, signaling, trafficking, etc.[1]

Glycan synthesis is a dynamic process that involves various enzymes involved in glycan attachment (glycosyltransferases), removal (glycosidases), glycan precursors synthesis and their amount, as well as sugar transporters delivering substrates for glycan chain synthesis[2]. In addition to enzymes directly involved in glycosylation, numerous other genetic loci have been implicated in the regulation of this complex process[3,4].

Tf N-glycans have been reported to be mainly diantennary and triantennary sialylated complex-type glycans, with or without fucose, with diantennary disialylated afucosylated glycans being the most abundant[5–8]. Aberrant Tf N-glycosylation has been predominantly studied in the context of a biomarker for congenital disorders of glycosylation[9–11], and chronic alcohol consumption[12], and has been seen in e.g. hepatocellular carcinoma[7], rheumatoid arthritis[13], inflammation[14]. Bergström et al. in 2008 reported small and mostly non-statistically significant differences in carbohydrate-deficient Tf patterns in 1387 individuals of different ethnicity, age, sex, and body mass index (BMI)[15], while observed differences in glycosylation profiles between smokers and non-smokers were attributed to higher alcohol intake in smokers. Besides that, Tf N-glycosylation has not been studied in much depth, and methods employed have usually been based on less sensitive and detailed techniques, e.g. lectins[16], isoelectric focusing[11], and high-performance liquid chromatography (HPLC) with absorbance detection (at 470 nm) of Tf glycoforms in an iron-Tf complex[15,17]. Recently, for Tf from a pooled healthy human serum, N-glycosylation has been described in more detail[8]. However, inter-individual variability or environmental factors influencing Tf N-glycosylation in a healthy human population remain largely unexplored, mainly because high-throughput methods enabling Tf purification and N-glycan analysis on a larger scale have been lacking.

One of the most studied glycoproteins is immunoglobulin G (IgG), predominantly because it is the most abundant antibody in human blood plasma, and one of the key molecules in the immune system response with N-glycosylation significantly impacting its function[18]. Moreover, there are efficient methods for its purification via binding to protein G or protein A. IgG N-glycans are predominantly of diantennary complex type, with a disialylated structure containing core fucose and bisecting N-acetylglucosamine (GlcNAc) being the most complex IgG N-glycan[19]. The variability of IgG N-glycosylation in the human population has been well studied[20–22] as well as its changes in different diseases[23].

Here, we present a highly specific robust high-throughput approach for purification and released N-glycan analysis of Tf from human blood plasma by ultra-high-performance liquid chromatography based on hydrophilic interactions and fluorescence detection (HILIC-UHPLC-FLD). We perform a detailed characterization of total Tf N-glycome by several complementary approaches—HILIC-UHPLC-FLD, exoglycosidase sequencing[24,25], and matrix-assisted laser desorption/ionization time-of-flight mass spectrometry (MALDI-TOF-MS) analysis after linkage-specific sialic acid ethyl esterification[26]. We describe Tf N-glycosylation variability within two healthy human populations expanding on our parallel genetic study performed on the same set of samples[27]. We demonstrate that Tf N-glycosylation is associated with age and sex, as well as with a number of biochemical and physiological traits. In parallel, we analyze IgG N-glycosylation in the same two populations and demonstrate that Tf N-glycosylation correlates more with sex than age compared to IgG N-glycosylation. Additionally, regression analysis with biochemical and physiological traits reveals different patterns of associations for Tf and IgG N-glycomes, supporting N-glycan importance in distinct physiological functions of these two glycoproteins.

## Results

**Highly specific high-throughput isolation of human Tf.** Tf was isolated in a high-throughput mode by immunoaffinity purification from blood plasma in two human populations, Korcula ($N = 927$, discovery cohort) and VIKING ($N = 958$, replication cohort) (Table 1), using our previously developed monolithic CIMac-@Tf 96-well plate with immobilized antibodies for human Tf[28]. Average purification capacity of the CIMac-@Tf 96-well plate was 300 μg of Tf per well (coefficient of variation, CV = 9.1% for the plate)[28]. Tf purification repeatability within the cohorts was determined from the standard plasma samples and resulted in a CV of 18.9% (average Tf mass = 156 ± 30 μg, $n = 72$) for Korcula and 23.4% (average Tf mass = 85 ± 20 μg, $n = 79$) for VIKING. In parallel with Tf isolation, IgG isolation ($N = 950$ and $N = 1087$ for Korcula and VIKING, respectively) and subsequent N-glycosylation profiling of both glycoproteins were done for the same individuals to allow their comparative analysis. Both Tf and IgG were deglycosylated with PNGase F, and their total released N-glycans labeled with 2-aminobenzamide (2-AB), cleaned up, and analyzed by HILIC-UHPLC-FLD (Fig. 1a). Since IgG was shown in our early experiments to be the major glycosylated contaminant in Tf eluate, first IgG was isolated from plasma samples using protein G 96-well monolithic plate, and flow-through was immediately applied on a CIMac-@Tf 96-well monolithic plate, thus bringing potential contamination from IgG in Tf eluate to a minimum (Supplementary Fig. 1a). Tf isolation process and purity of isolated Tf have been controlled throughout the study by sodium dodecyl sulfate–polyacrylamide electrophoresis (SDS–PAGE) analysis of a blank sample and four randomly taken Tf eluates from each plate, demonstrating successful

**Table 1 Demographic characteristics of Korcula and VIKING cohorts.**

| Characteristic | Korcula, $N = 927^a$ | VIKING, $N = 958^a$ |
|---|---|---|
| Age | 55 (41-67) | 52 (41-63) |
| Sex | | |
| F | 600 (65) | 578 (60) |
| M | 327 (35) | 380 (40) |
| BMI | 26.6 (23.7-29.4) | 26.8 (24.2-29.9) |
| Smoking | | |
| Never smoked | 274 (30) | 517 (54) |
| Stopped | 434 (48) | 368 (38) |
| Currently | 196 (22) | 72 (7.5) |
| PerStill | | |
| No | 334 (57) | 326 (56) |
| Yes | 253 (43) | 251 (44) |
| Cholesterol | 5.70 (5.00-6.70) | 5.30 (4.60-6.00) |
| Triglycerides | 1.20 (0.90-1.60) | 0.90 (0.60-1.30) |
| HDL | 1.50 (1.30-1.70) | 1.47 (1.24-1.76) |
| LDL | 3.65 (2.90-4.40) | 3.32 (2.71-4.00) |
| Insulin | NA | 6.2 (4.3-9.0) |
| HbA1c | 5.30 (5.00-5.60) | 5.30 (5.20-5.50) |

*F* females, *M* males, *BMI* body mass index, *PerStill* having a regular menstrual cycle, *HDL* high-density lipoprotein, *LDL* low-density lipoprotein, *HbA1c* hemoglobin A1c, *IQR* interquartile range.
$^a$Median (IQR); N (%).

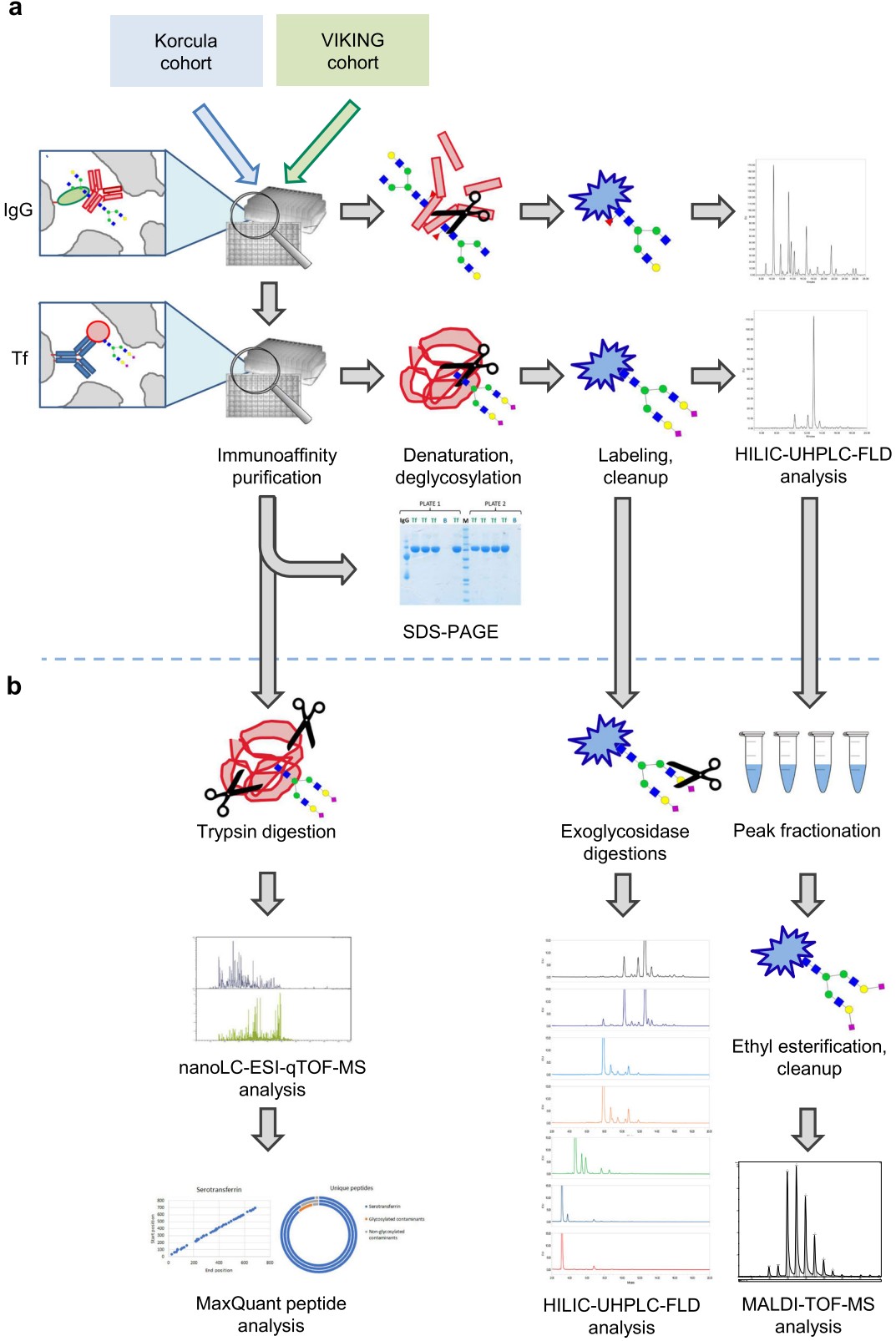

**Fig. 1 Workflow for human plasma transferrin (Tf) isolation and N-glycan structures characterization. a** Sequential isolation and N-glycan profiling of immunoglobulin G (IgG) and Tf from two human populations (Korcula and VIKING). HILIC-UHPLC-FLD ultra-high-performance liquid chromatography based on hydrophilic interactions with fluorescent detection. **b** Workflow for Tf purity assessment and N-glycan structures characterization by several complementary approaches. nanoLC-ESI-qTOF-MS nano-liquid chromatography coupled to electrospray ionization quadrupole time-of-flight mass spectrometry; MALDI-TOF-MS matrix-assisted laser desorption/ionization time-of-flight mass spectrometry.

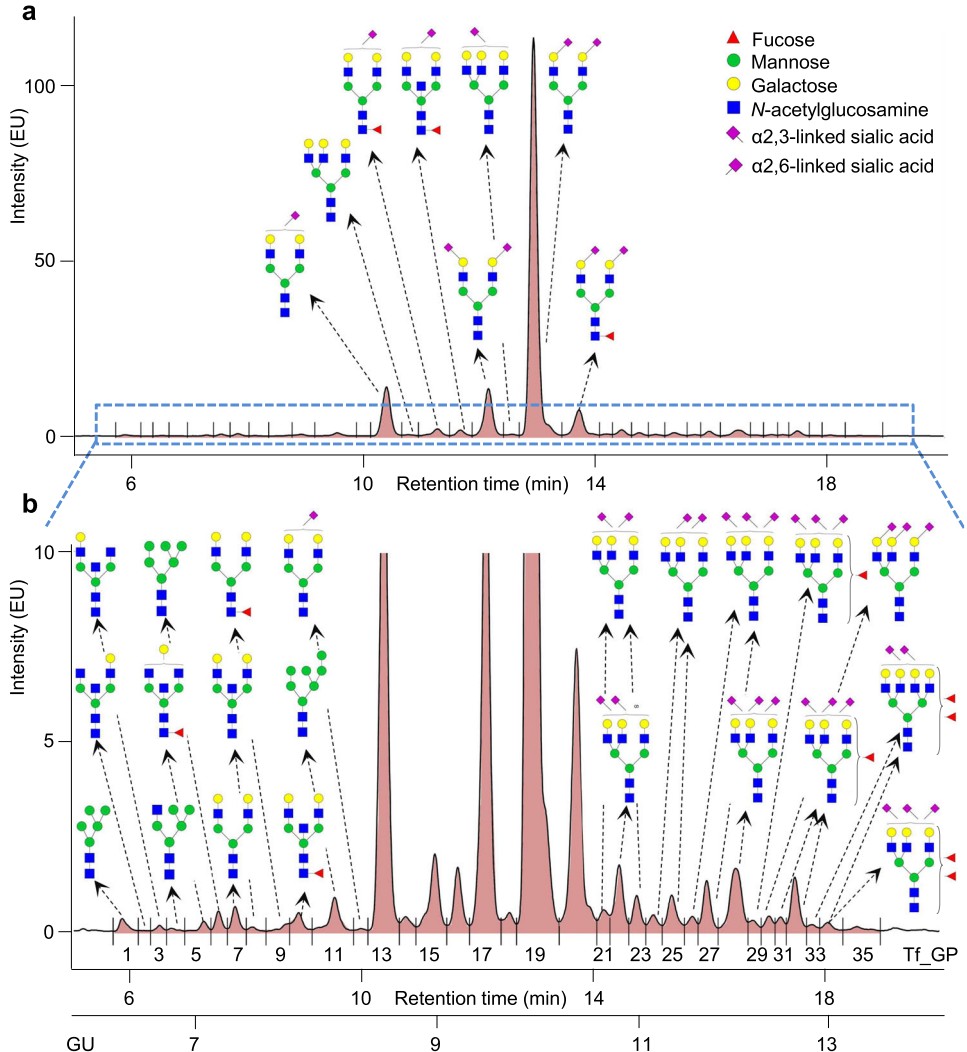

**Fig. 2 N-glycosylation profile of human plasma transferrin (Tf) obtained by ultra-high-performance liquid chromatography based on hydrophilic interactions with fluorescent detection (HILIC-UHPLC-FLD).** The most abundant glycan structure in each Tf glycan peak (Tf_GP) is shown. For full characterization see Supplementary Table 2. Sialic acid linkages are assigned based on matrix-assisted laser desorption/ionization time-of-flight mass spectrometry (MALDI-TOF-MS) analysis of fractionated 2-aminobenzamide (2-AB) labeled and ethyl-esterified N-glycans eluting in individual Tf_GPs. Structural schemes are given according to Consortium for Functional Glycomics (CFG) guidelines. EU emission unit, GU glucose unit. **a** The eight most abundant Tf N-glycan structures are shown. **b** Zoomed-in view showing less abundant Tf_GPs.

Tf isolation and no visible presence of other protein contaminants (Supplementary Fig. 1b). Moreover, we wanted to ensure that quantified released N-glycans indeed originated from Tf, and not other potentially glycosylated contaminants present in the amount below SDS–PAGE detection. Therefore, isolated Tf purity was further assessed by proteomic analysis of Tf eluate (Supplementary Table 1). Proteomic analysis of Tf eluates resulted in the average relative intensity extracted for serotransferrin (UniProt P02787) of 99.36%, confirming the high purity of the Tf sample used for subsequent released N-glycan analysis. Moreover, other glycosylated contaminants were detected in only one replicate with a relative intensity of 0.22% (Supplementary Table 1) additionally increasing the confidence that released N-glycans originate almost exclusively from Tf.

**Characterization of human plasma Tf N-glycans.** Human plasma Tf N-glycome was analyzed on the level of released glycans by HILIC-UHPLC-FLD and separated into 35 glycan peaks (Tf_GP1-Tf_GP35, Fig. 2, Supplementary Table 2). Several complementary approaches have been used to structurally

characterize N-glycans eluting in each Tf_GP (Fig. 1b). Preliminary structures were assigned according to glucose unit (GU) values and GlycoStore database search (www.glycostore.org)[29,30], and additionally confirmed by exoglycosidase sequencing[24,25] (Supplementary Table 3, Supplementary Figs. 2 and 3) and MALDI-TOF-MS analysis of fractionated 2-AB labeled N-glycans eluting in individual Tf_GPs. MALDI-TOF-MS analysis was performed with or without ethyl esterification of collected N-glycan fractions, which enabled differentiation of α2,3- and α2,6-bound sialic acid[26]. N-glycan structures were successfully assigned to 34/35 Tf_GPs accounting for approximately 99.8% of Tf N-glycome (Fig. 2, Supplementary Table 2).

Predominantly, one glycan structure has been detected per chromatographic peak ranging in their complexity from high-mannose glycans, e.g. M5 in Tf_GP1, to complex tetraantennary sialylated and fucosylated glycans, e.g. F2A4G4S[3,3]2 in Tf_GP33. The most abundant glycans in Tf N-glycome are of diantennary digalactosylated type with two α2,6-sialic acids without (A2G2S[6,6]2, Tf_GP19) or with core fucose (FA2G2S[6,6]2, Tf_GP20), with both α2,3- and α2,6-sialic acid

without core fucose (A2G2S[3,6]2, Tf_GP17), and α2,6-monosialylated structure without core fucose (A2G2S[6]1, Tf_GP13). In addition to directly measured N-glycans (Fig. 2, Supplementary Table 2, Supplementary Fig. 4), derived glycosylation traits that are biologically more related to activities of specific enzymes in the glycosylation pathway were calculated for both the Tf N-glycome (Supplementary Table 4) and IgG N-glycome (Supplementary Table 5).

The whole sample analysis procedure (from IgG and Tf isolation to HILIC-UHPLC-FLD analysis) was controlled by having one negative control (blank) and four to six internal standard samples per plate, according to which the method variability was assessed (Supplementary Table 6/Supplementary Figs. 5 and 6). Korcula cohort additionally contained nine to ten sample duplicates per each 96-well plate for which Tf N-glycan analysis showed a very good correlation (Supplementary Table 6/Supplementary Fig. 7).

**Age- and sex-dependent differences in the Tf N-glycome**. To determine to which extent Tf N-glycome was associated with age and sex, regression analysis was performed in two independent cohorts (Table 1). After correction for multiple testing, statistically significant associations with both age and sex were observed for a majority of glycan structures (Supplementary Tables 7 and 8). These associations replicated well between the two populations, though some population-specific differences existed (Fig. 3). The most considerable associations with age were observed in the levels of glycan branching, galactosylation and sialylation. Triantennary (A3) and tetraantennary (A4) glycans, trigalactosylated (G3) and tetragalactosylated (G4) glycans, disialylated (S2) and trisialylated (S3) glycans steadily increased with age. On the other hand, diantennary (A2) glycans, agalactosylated (G0), monogalactosylated (G1), and digalactosylated (G2) glycans, asialylated (S0) and monosialylated (S1) glycans, high-mannose glycans (HM) and glycans with bisecting GlcNAc (B) or fucose (F) decreased with age. The observed changes were present both in men and women (Supplementary Table 9), with patterns of change through time being similar (Fig. 3).

Regression analysis revealed considerable differences in Tf N-glycome between females and males (Supplementary Tables 7 and 8). The largest differences were observed in the levels of bisecting GlcNAc, sialylation, and galactosylation. Statistically significant increases in S2, G2 and G4, A2 and A4 glycans were observed in males. Females showed higher levels of HM glycans, glycans with B or F, G0 and G1, S0 and S1 glycans (Supplementary Table 7). These differences replicated well between the two populations.

While a number of associations between Tf N-glycome and age were observed, with 30 out of 35 Tf GPs being significantly associated with age (Supplementary Table 8), the strongest associations were not as strong as associations observed between IgG N-glycome and age (Supplementary Fig. 8). The strongest associations between Tf N-glycome and age were observed for A4 ($R = 0.278$, $p.adj = 6.87E-39$), G4 ($R = 0.278$, $p.adj = 6.87E-39$), and S1 ($R = -0.262$, $p.adj = 9.30E-34$) glycans for which <10% of the variance was explained by age (Supplementary Table 10), compared to the strongest associations between IgG N-glycome and age, for G0 ($R = 0.666$, $p.adj < 1.00E-300$), G2 ($R = -0.667$, $p.adj < 1.00E-300$) and S1 ($R = -0.595$, $p.adj < 1.00E-300$) glycans for which more than 40% of the variance was explained by age (Supplementary Table 11). In contrast to age-related associations, associations between sex and Tf N-glycome were stronger than those with IgG N-glycome. For the strongest observed association between Tf N-glycome and sex, B glycans ($R = -0.331$, $p.adj = 3.63E-57$), more than 10% of the variation

was explained by sex (Supplementary Table 10); while for the strongest observed association between IgG N-glycome and sex, G2 glycans ($R = -0.209$, $p.adj = 1.84E-22$), <5% of the variation was explained by sex (Supplementary Table 11). Patterns of observed correlations revealed that age and sex have quite different influences on Tf and IgG N-glycomes. For the IgG N-glycome, the strongest observed age effects were several times stronger than the strongest observed sex effects. Unlike for IgG N-glycome, the strongest observed age and sex associations for Tf N-glycome were of similar magnitude.

**Tf N-glycome and biochemical and physiological traits**. To identify factors that may be responsible for the remaining variability in the Tf N-glycome, we performed regression analysis with available biochemical and physiological traits in our databases (Supplementary Table 12). Since the majority of available biochemical and physiological traits correlate with age and sex and consequently univariate regression analysis showed significant associations for virtually all traits (Supplementary Fig. 9), age and sex were included as covariates in all further analyses (Fig. 4). Regression analysis revealed strong associations between multiple Tf N-glycome and clinical (weight, BMI, blood pressure, etc.) as well as biochemical (HDL, insulin, triglycerides, etc.) parameters (Supplementary Tables 13 and 14). These associations replicated well between the two populations. The strongest associations were observed between Tf N-glycome and weight, with 22 out of 35 Tf GPs being significantly associated with weight (Supplementary Table 14). A2, S2, and G2 glycans were positively correlated with weight. A3, G1, and G3 glycans, S0 and S1 glycans, B glycans, and HM glycans were negatively correlated with weight (Supplementary Table 13). The majority of clinical and biochemical traits that are known to be associated with an unhealthy lifestyle (insulin, uric acid, glucose, triglycerides, HbA1c, blood pressure) revealed patterns of correlations with Tf N-glycome similar to those obtained for weight. A similar pattern was observed for several medical conditions (hypertension, gout, diabetes), although associations were not as strong as for biochemical traits (Supplementary Table 13). Since the majority of biochemical and physiological traits correlate with weight, the possibility remained that some of the observed associations were a reflection of weight. To further investigate the impact biochemical and physiological traits have on Tf N-glycome, regression analysis was repeated with BMI included as an additional covariate (Table 2, Supplementary Tables 15 and 16). While the number of significant associations in the repeated analysis was lower, with obtained effects not as strong as effects from previous analysis with only age and sex as covariates, the pattern of correlations with Tf N-glycome remained similar (Supplementary Fig. 10). Most associations between clinical parameters, medical conditions, and Tf N-glycome were reduced below the level of statistical significance, however, most of the associations with biochemical parameters remained statistically significant (Supplementary Tables 15 and 16).

Regression analysis with biochemical and physiological traits revealed different patterns of associations for Tf and IgG N-glycomes (Supplementary Tables 17–20). For most clinical parameters (BMI, weight, blood pressure), medical conditions (hypertension, gout, diabetes, arthritis), and biochemical parameters (insulin, HDL, uric acid, glucose, triglycerides, HbA1c, fibrinogen) considerably stronger associations were observed with Tf N-glycome in comparison with IgG N-glycome. For IgG N-glycome, the analysis showed comparatively stronger associations only for traits related to fecundity (regular menstrual cycle) and smoking, although significant associations between Tf N-glycome and smoking were also observed.

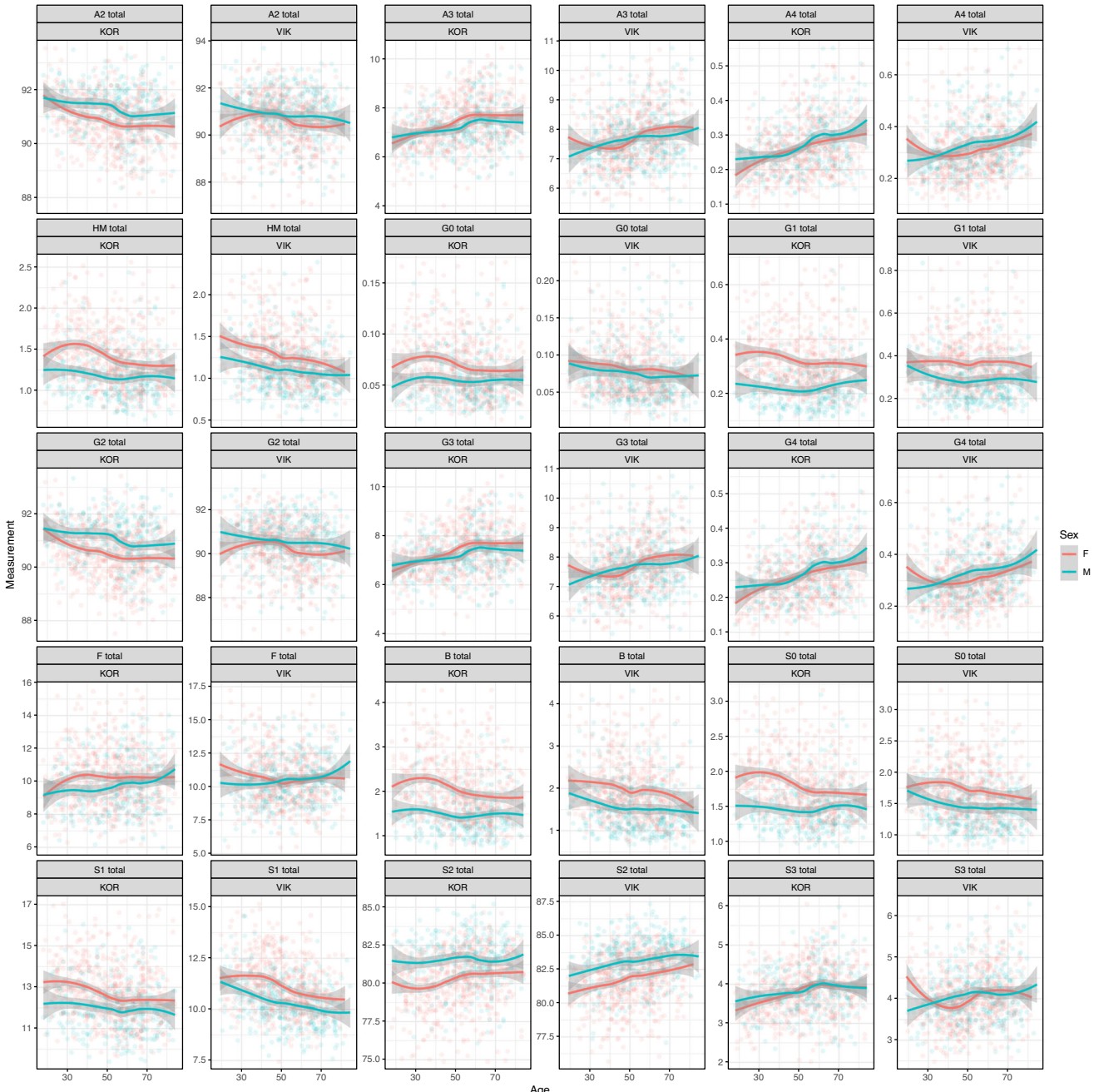

**Fig. 3 N-glycosylation profile of human plasma transferrin (Tf) is associated with age and sex.** Derived N-glycosylation traits are shown for each cohort (KOR—Korcula, VIK—VIKING) and calculated according to formulas in Supplementary Table 4. Turquoise and red curves are fitted local regression models describing a sex-specific relationship between age and derived traits. The gray-shaded region is a pointwise 95% confidence interval on the fitted values (there is 95% confidence that the true regression curve lies within the shaded region). F females, M males, A2 diantennary glycans, A3 triantennary glycans, A4 tetraantennary glycans, HM high-mannose glycans, G0 agalactosylated glycans, G1 monogalactosylated glycans, G2 digalactosylated glycans, G3 trigalactosylated glycans, G4 tetragalactosylated glycans, F glycans containing fucose, B glycans containing bisecting N-acetylglucosamine, S0 asialylated glycans, S1 monosialylated glycans, S2 disialylated glycans, S3 trisialylated glycans.

## Discussion

Glycosylation is one of the most common co- and posttranslational modifications affecting not only protein structure but also its functionality[1]. Variability of IgG glycosylation has been extensively studied in different populations[20–22], diseases[23], and functional studies[18,31]. However, glycosylation changes of other glycoproteins, including Tf, have been mostly studied in the context of changes happening in different disease states vs. healthy controls, while their variability in a healthy human population remains generally unknown and is considered stable.

Moreover, while high-throughput purification (and glycosylation analysis) strategies for large-scale study applications do exist for IgG, this is largely not the case for other glycoproteins. For only a few other glycoproteins, e.g. IgA[32], apolipoprotein CIII[33], α-1-acid glycoprotein[34,35], high-throughput purification and/or glycosylation analysis strategies with application in population studies have been reported.

Here, we have developed a highly specific robust high-throughput approach for purification and glycosylation analysis of Tf from human blood plasma by HILIC-UHPLC-FLD.

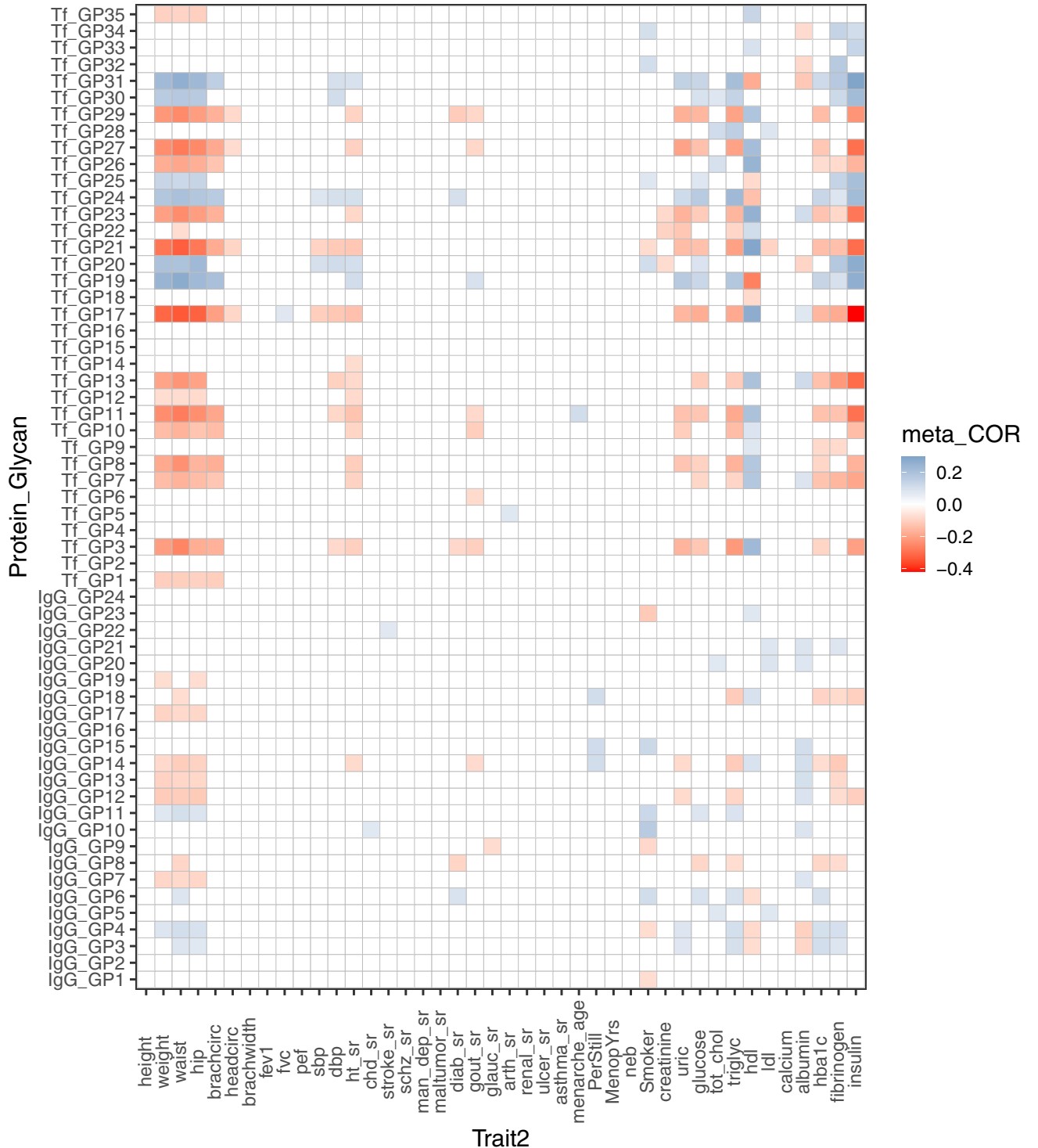

**Fig. 4 Age- and sex-adjusted correlations between directly measured transferrin (Tf)/immunoglobulin G (IgG) N-glycan traits (GPs) and biochemical and physiological traits.** Correlation analysis was performed on Korcula and VIKING cohorts separately and then combined using an inverse-variance weighted meta-analysis approach. Prior to correlation analysis, glycan measurements were adjusted for age and sex. Only statistically significant (adjusted meta *p*-value < 0.01, method = Benjamini–Hochberg) meta-correlations are shown, where the blue color indicates a positive correlation, and the red color indicates a negative correlation. Description of biochemical and physiological traits is given in Supplementary Table 12.

Developed reusable monolithic CIMac-@Tf 96-well plate with immobilized antibodies for human Tf enables fast large-scale and cost-effective purification of Tf without the use of packed columns or tips used in previously reported approaches[8,36]. Combining the developed Tf purification workflow with N-glycan quantification by HILIC-UHPLC-FLD enabled a detailed

characterization of total released Tf N-glycome, including information on sialic acid linkages using ethyl esterification and MALDI-TOF-MS analysis[26] of fractionated chromatographic peaks. The repertoire of identified Tf N-glycan structures expands on the previous work[8,37], especially in terms of sialic acid linkages for disialylated and trisialylated structures, making this study the

**Table 2 Age-, sex- and BMI-adjusted associations of derived transferrin (Tf) N-glycome traits with biochemical and physiological traits.**

| Glycan | Trait | Korcula | | | | | VIKING | | | | | META | | | |
|---|---|---|---|---|---|---|---|---|---|---|---|---|---|---|---|
| | | DF | effect | SE | p.val | p.adj | DF | effect | SE | p.val | p.adj | effect | SE | p.val | p.adj |
| S2 total | insulin | NA | NA | NA | NA | NA | 948 | 0.248 | 0.034 | 7.02E−13 | 4.00E−10 | 0.248 | 0.034 | 4.53E−13 | 2.86E−10 |
| S1 total | insulin | NA | NA | NA | NA | NA | 948 | −0.234 | 0.034 | 7.34E−12 | 2.09E−09 | −0.234 | 0.034 | 5.22E−12 | 1.64E−09 |
| G2 total | hdl | 865 | −0.203 | 0.035 | 4.63E−09 | 2.03E−06 | 945 | −0.114 | 0.037 | 1.82E−03 | 4.31E−02 | −0.162 | 0.025 | 1.38E−10 | 2.91E−08 |
| A2 total | hdl | 865 | −0.204 | 0.035 | 6.59E−09 | 2.03E−06 | 945 | −0.112 | 0.037 | 2.27E−03 | 4.61E−02 | −0.160 | 0.025 | 2.60E−10 | 4.10E−08 |
| A3 total | hdl | 865 | 0.181 | 0.036 | 4.59E−07 | 7.05E−05 | 945 | 0.127 | 0.036 | 4.80E−04 | 1.83E−02 | 0.155 | 0.026 | 1.51E−09 | 1.58E−07 |
| G3 total | hdl | 865 | 0.181 | 0.036 | 4.59E−07 | 7.05E−05 | 945 | 0.127 | 0.036 | 4.80E−04 | 1.83E−02 | 0.155 | 0.026 | 1.51E−09 | 1.58E−07 |
| S1 total | fbrinogen | 846 | −0.103 | 0.033 | 1.57E−03 | 2.42E−02 | 945 | −0.151 | 0.032 | 2.01E−06 | 2.87E−04 | −0.128 | 0.023 | 2.04E−08 | 1.84E−06 |
| S2 total | hdl | 865 | −0.106 | 0.035 | 2.20E−03 | 2.98E−02 | 945 | −0.150 | 0.033 | 7.20E−06 | 6.84E−04 | −0.129 | 0.024 | 8.92E−08 | 7.03E−06 |
| S2 total | hba1c | 860 | 0.158 | 0.035 | 5.70E−06 | 3.90E−04 | 946 | 0.086 | 0.034 | 1.16E−02 | 1.27E−01 | 0.121 | 0.024 | 6.53E−07 | 4.57E−05 |
| S0 total | insulin | NA | NA | NA | NA | NA | 948 | −0.178 | 0.036 | 9.24E−07 | 1.76E−04 | −0.178 | 0.036 | 9.00E−07 | 5.67E−05 |
| S3 total | tot_chol | 865 | 0.109 | 0.036 | 2.68E−03 | 3.50E−02 | 948 | 0.124 | 0.035 | 3.39E−04 | 1.76E−02 | 0.117 | 0.025 | 3.25E−06 | 1.86E−04 |
| F total | fibrinogen | 846 | 0.109 | 0.035 | 1.84E−03 | 2.76E−02 | 945 | 0.120 | 0.035 | 6.94E−04 | 2.24E−02 | 0.114 | 0.025 | 4.48E−06 | 2.35E−04 |
| S3 total | hdl | 865 | 0.142 | 0.037 | 1.22E−04 | 5.34E−03 | 945 | 0.094 | 0.037 | 1.06E−02 | 1.23E−01 | 0.118 | 0.026 | 6.38E−06 | 3.09E−04 |
| HM total | insulin | NA | NA | NA | NA | NA | 948 | −0.162 | 0.036 | 6.99E−06 | 6.84E−04 | −0.162 | 0.036 | 6.99E−06 | 3.15E−04 |
| S0 total | triglyc | 865 | −0.094 | 0.035 | 6.57E−03 | 6.12E−02 | 945 | −0.118 | 0.034 | 4.61E−04 | 1.83E−02 | −0.106 | 0.024 | 1.13E−05 | 4.73E−04 |
| A4 total | fibrinogen | 846 | 0.114 | 0.033 | 4.84E−04 | 1.24E−02 | 945 | 0.087 | 0.035 | 1.21E−02 | 1.28E−01 | 0.102 | 0.024 | 2.14E−05 | 7.95E−04 |
| G4 total | fibrinogen | 846 | 0.114 | 0.033 | 4.84E−04 | 1.24E−02 | 945 | 0.087 | 0.035 | 1.21E−02 | 1.28E−01 | 0.102 | 0.024 | 2.14E−05 | 7.95E−04 |
| G2 total | uric | 865 | 0.192 | 0.040 | 1.65E−06 | 2.03E−04 | 948 | 0.045 | 0.041 | 2.68E−01 | 6.74E−01 | 0.120 | 0.029 | 2.65E−05 | 9.27E−04 |
| S1 total | hba1c | 860 | −0.129 | 0.036 | 3.41E−04 | 1.05E−02 | 946 | −0.077 | 0.034 | 2.15E−02 | 1.81E−01 | −0.102 | 0.025 | 3.92E−05 | 1.30E−03 |
| G1 total | triglyc | 865 | −0.093 | 0.034 | 5.94E−03 | 5.89E−02 | 945 | −0.105 | 0.034 | 2.21E−03 | 4.61E−02 | −0.099 | 0.024 | 4.21E−05 | 1.33E−03 |
| G1 total | arth_sr | 845 | 0.380 | 0.130 | 3.44E−03 | 4.25E−02 | 947 | 0.316 | 0.115 | 5.76E−03 | 9.12E−02 | 0.344 | 0.086 | 6.54E−05 | 1.96E−03 |
| A2 total | uric | 865 | 0.187 | 0.041 | 3.98E−06 | 3.18E−04 | 948 | 0.040 | 0.041 | 3.25E−01 | 7.28E−01 | 0.115 | 0.029 | 7.04E−05 | 2.02E−03 |
| S0 total | waist | 867 | −0.120 | 0.060 | 4.52E−02 | 2.53E−01 | 948 | −0.250 | 0.071 | 3.99E−04 | 1.83E−02 | −0.175 | 0.046 | 1.40E−04 | 3.76E−03 |

All associations are reported in Supplementary Table 15. Description of biochemical and physiological traits is given in Supplementary Table 12.
*DF* degrees of freedom, *SE* standard error, *p.adj* false discovery rate was controlled using Benjamini–Hochberg method at the specified level of 0.05, *S2* disialylated glycans, *S1* monosialylated glycans, *S0* asialylated glycans, *S3* trisialylated glycans, *G2* digalactosylated glycans, *G3* trigalactosylated glycans, *G4* tetragalactosylated glycans, *HM* high-mannose glycans, *A4* tetraantennary glycans, *A2* diantennary glycans, *A3* triantennary glycans, *G1* monogalactosylated glycans.

most extensive one in terms of Tf N-glycome characterization on the level of total released N-glycans.

Previously, Bergström et al. studied differences in Tf glycosylation patterns related to ethnicity, age, sex, BMI, and smoking as potential confounders for carbohydrate-deficient Tf testing in the context of a biomarker for heavy alcohol use[15]. Based on the relative quantification of seven iron-saturated Tf glycoforms (asialo-, monosialo-, disialo-, trisialo-, tetrasialo-, pentasialo-, and hexasialotransferrin) in serum using an HPLC[17] they concluded that there are no significant differences in disialotransferrin levels (the main carbohydrate-deficient Tf glycoform) between individuals of different ethnic origin, age, sex, BMI or smoking status[15]. They did observe some statistically significant differences in other Tf glycoforms between individuals of different sex, BMI, and smoking statuses, although they concluded that the latter mostly reflected higher alcohol consumption by smokers.

We have applied the developed state-of-the-art analytical approach to explore in-depth the variability of Tf N-glycome in a healthy human population and to compare it with the variability of IgG N-glycosylation in the same set of samples. By relative quantification of 35 directly measured and 15 calculated derived Tf N-glycosylation traits, we performed the most extensive analysis of Tf N-glycome variability in a healthy human population demonstrating that Tf N-glycome associates with the age and sex of individuals, as well as with a number of biochemical and physiological traits. Compared to the IgG glycome, Tf N-glycome correlates more with sex than age, which could be a reflection of the different roles of glycans on these two proteins. As the main function of Tf is the binding and transport of iron it is somewhat expected that its glycosylation is well conserved during aging, while on the other hand, it is plausible that its synthesis and glycosylation are differentially regulated in females compared to males due to, e.g. menstrual cycles and hormonal fluctuations[13]. It is known that both Tf concentration and glycosylation change during pregnancy[13,38,39] as well as during the use of oral contraceptives[13]. Although the biological meaning of different Tf glycosylation patterns is still not completely understood, early research studies have demonstrated that the absence of Tf glycosylation doesn't affect iron binding or Tf binding to a Tf receptor[40–42], while it was shown that it affects cellular iron uptake in vitro and in vivo[40,42]. However, changes in Tf N-glycome can alter its circulation half-life and affect iron transport[43], which might be an important element of the response to acute inflammation[44,45]. On the other hand, the function of IgG is recognition of foreign pathogens and activation of an appropriate immune response, so it is conceivable IgG glycosylation correlates more with age than with sex, since it was proposed that it somewhat reflects previous antigen encounters during past infections[46–48].

The role of IgG N-glycans in the modulation of its effector functions has been well recognized[49–52]. One of the most studied examples is the addition of core fucose to the Fc fragment Asn297 N-glycosylation site that decreases antibody-dependent cell-mediated cytotoxicity (ADCC) up to 100-fold through lower IgG affinity toward Fcγ receptor IIIa[53,54]. Moreover, both galactosylation and sialylation have been shown to modulate the inflammatory activity of IgG. While galactosylated IgG has an anti-inflammatory effect by inhibiting the complement pathway[55], agalactosylated IgG glycans act pro-inflammatory by activating the alternative pathway[56], as well as lectin complement pathway via interaction with mannose-binding lectin[57]. Sialylation also changes IgG activity from pro-inflammatory to anti-inflammatory[31,51], although the exact mechanisms remain to be elucidated. Pro-inflammatory glycosylation profile of IgG, predominantly characterized by decreased galactosylation and sialylation, has been seen with aging[21,58], as well as with different

diseases[23]. It was hypothesized that the lifelong antigenic challenges and increasing antigenic burden cause immune system remodeling leading to the state of a low-grade chronic inflammation that characterizes aging, i.e. inflammaging[46]. In this context, pro-inflammatory IgG glycans are considered both biomarkers and functional effectors of aging[47].

Next, we demonstrated that Tf N-glycome correlates with a number of biochemical and physiological traits and with different patterns of association compared to the IgG N-glycome validating distinctive protein-specific roles of N-glycans in Tf and IgG physiological functions. Strong associations of Tf N-glycome with weight and other clinical and biochemical parameters potentially implicate systemic changes in the iron transport by Tf within the organism, e.g. after intestinal iron uptake, by iron recycling from iron-containing proteins or iron stores[59], reflected by changes in these physiological and metabolic factors. During iron transport, Tf binds to two plasma membrane receptors—Tf receptor and asialoglycoprotein receptor, with early studies demonstrating that the affinity of asialo Tf glycoforms for asialoglycoprotein receptor differs depending on the Tf glycan chain complexity[60]. Even though the functional relevance of Tf glycosylation remains underexplored, it is possible that specific Tf glycoforms influence Tf affinity and interactions with its receptors, fine-tuning the iron transport between sites of absorption and delivery to requiring cells[59].

In conclusion, by developing a robust high-throughput approach for the analysis of total Tf N-glycome we have expanded the current knowledge of human plasma Tf N-glycan repertoire. Moreover, we have performed an extensive analysis of Tf N-glycome variability in a healthy human population demonstrating its changes with age, sex, biochemical, and physiological status of individuals, providing the basis for future population and clinical studies. By performing IgG glycosylation analysis in the same samples, we demonstrated that glycosylation of these two proteins has independent regulation, which confirms a similar observation from a previous genetic study[27].

## Methods

### Sample collection

*Korcula cohort.* This study was performed in the adult population of the island of Korčula, Croatia[61]. The fieldwork was performed in 2007 in the eastern parts of the island, focusing on the town of Korčula and villages Lumbarda, Žrnovo, and Račišće. The sampling approach was convenient, with population-wide invites to non-institutionalized individuals living in the island. All subjects were aged 18 and over, and had signed informed consent before entering the study, which was approved by the Ethical Committee of the Medical School, University of Zagreb, and Multi-Centre Research Ethics Committee for Scotland. All relevant ethical regulations were followed. Fasting plasma samples were collected, processed, and stored at −80 °C on-site immediately upon processing, ensuring the highest possible sample quality[62].

*VIKING cohort.* The Viking Health Study—Shetland (VIKING) is a family-based, cross-sectional study that seeks to identify genetic factors affecting cardiovascular and other disease risks in the isolated population of the Shetland Islands in northern Scotland[63]. Compared to Mainland Scotland, genetic diversity in this population is decreased, consistent with the high levels of endogamy historically. Between 2013 and 2015, 2105 participants were recruited, having at least three grandparents from Shetland. Numerous health-related phenotypes and environmental exposures were measured and fasting blood samples were collected from each individual. The study was approved by the South East Scotland Research Ethics Committee, NHS Lothian (reference: 12/SS/0151), all participants gave informed consent, and all relevant ethical regulations were followed.

**Experimental design**. This is an observational study and samples were collected as samples of convenience. No statistical calculation of sample size was performed; the sample size was determined based on availability. All samples were processed in batches (96-well in the case of Korcula samples and 60-well in the case of VIKING samples), following a predetermined experimental design, which was blocked on sex and age information. All plates included several technical replicates of a standard plasma sample and blank samples for quality control and batch correction. Standard plasma sample used for Korcula analysis was prepared as a pool of

cohort samples. Standard plasma sample used for VIKING cohort analysis was obtained from the Croatian National Institute of Transfusion Medicine after approval by the Ethical Committee of the institute. Additionally, Korcula samples contained around 10% of duplicate samples per plate to further assess method variability. The sample analysis was done blindly, and all samples were analyzed with a uniform set of techniques.

**Immunoglobulin G and transferrin isolation**. The whole procedure was done in a 96-well plate manner according to the previously prepared randomization plan and ultrapure water (≥18.2 MΩ at 25 °C) was used throughout. IgG was isolated from blood plasma samples by CIM® Protein G 96-well monolithic plate (BIA Separations, a Sartorius company, Ajdovščina, Slovenia) with individual column volume of 200 µL using a vacuum manifold (Pall Corporation, Ann Arbor, MI, USA)[20]. All steps during the isolation procedure were performed at around 380 mmHg, except for plasma sample application and glycoprotein elution (around 200 mmHg). The protein G plate was preconditioned by washing with 2 mL of ultrapure water, 2 mL of 1× PBS, pH 7.4, 1 mL of 0.1 mol L$^{-1}$ formic acid (Merck, Darmstadt, Germany), 2 mL of 10× PBS, and 4 mL of 1× PBS, pH 7.4. Then, 120 µL of plasma sample was centrifuged for 3 min at 12,100×g, diluted with 1× PBS, pH 7.4 (1:7), and filtered through a 0.45 µm AcroPrep hydrophilic polypropylene (GH Polypro, GHP) filter plate (Pall Corporation) using a vacuum manifold (around 380 mmHg). After plasma filtration, samples were applied to the preconditioned protein G plate and flowthrough was collected for immediate subsequent Tf isolation by previously developed CIMac-@Tf 96-well monolithic plate (BIA Separations)[28] with individual column volume of 200 µL. The CIMac-@Tf 96-well monolithic plate was preconditioned by washing with 2 mL of ultrapure water, 1 mL of 0.1 mol L$^{-1}$ formic acid pH 3.0 (pH adjusted with 25% ammonia solution, Merck), and 4 mL of 1× PBS, pH 7.4. Unbound proteins during IgG and Tf isolation were washed away with 6 mL of 1× PBS, pH 7.4, and 1× PBS (0.25 mol L$^{-1}$ NaCl), pH 7.4, respectively. Bound IgG was eluted with 1 mL of 0.1 mol L$^{-1}$ formic acid and bound Tf with 0.7 mL of 0.1 mol L$^{-1}$ formic acid pH 3.0 and immediately neutralized with 1 mol L$^{-1}$ ammonium hydrogencarbonate (Sigma-Aldrich, St. Louis, MO, USA) to pH 7.0. The protein G plate was regenerated by washing with 2 mL of 0.1 mol L$^{-1}$ formic acid, 2 mL of 10× PBS, 4 mL of 1× PBS, pH 7.4, and 1 mL of 20% (v/v) ethanol in 20 mmol L$^{-1}$ Tris + 0.1 mol L$^{-1}$ NaCl, while the CIMac-@Tf plate was washed with 2 mL of 0.1 mol L$^{-1}$ formic acid pH 3.0, 4 mL of 1× PBS, pH 7.4, and 1 mL of 1× PBS + 0.02% NaN$_3$ (Sigma-Aldrich), pH 7.4. Monolithic plates were stored in 20% (v/v) ethanol in 20 mmol L$^{-1}$ Tris + 0.1 mol L$^{-1}$ NaCl (protein G plate) and 1× PBS + 0.02% NaN$_3$, pH 7.4 (CIMac-@Tf plate) at 4 °C until the next isolation. IgG and Tf concentrations were measured at 280 nm using a NanoDrop 8000 spectrophotometer (Thermo Fisher Scientific, Waltham, ME, USA). Each elution fraction (300 µL) was dried in a vacuum centrifuge (Thermo Fisher Scientific) and stored at −20 °C until subsequent N-glycan analysis.

**Sodium dodecyl sulfate–polyacrylamide gel electrophoresis**. Tf elution fractions were analyzed by sodium dodecyl sulfate–polyacrylamide gel electrophoresis (SDS–PAGE) using NuPAGE 4–12% Bis-Tris gradient protein gels (1.0 mm thickness) under reducing conditions according to the manufacturer's instructions (Invitrogen, Waltham, MA, USA). The gels were run at 200 V for 35 min using a NuPAGE MES SDS buffering system (Invitrogen). Protein bands were visualized by GelCode Blue staining reagent (Thermo Fisher Scientific).

**Transferrin purity assessment by proteomic analysis**. Tf purity after isolation was assessed by performing trypsin digestion and LC–MS analysis of obtained (glyco)peptides. Pooled isolated Tf sample (≈20 µg of protein in 150 µL of the eluate) was reduced with 16 µL of 20 mmol L$^{-1}$ dithiothreitol (Sigma-Aldrich), incubated at 60 °C for 30 min and then alkylated with the addition of 30 µL of 40 mmol L$^{-1}$ iodoacetamide (IAA, Sigma-Aldrich) and incubation in the dark for 30 min. Before overnight trypsin digestion (0.2 µg per sample), samples were left in bright light at room temperature for 20 min to stop IAA activity. Tryptic glycopeptides and peptides were separated and analyzed by nano-liquid chromatography (Waters, Milford, MA, USA) coupled to electrospray ionization quadrupole time-of-flight mass spectrometry (Bruker Daltonics, Bremen, Germany) (nanoLC-ESI-qTOF-MS). Samples were loaded on a PepMap 100 C18 trap column (5 mm × 300 µm i.d.; Thermo Fisher Scientific) for 3 min of trapping at a 40 µL min$^{-1}$ flow rate. After that, the trap column was switched in-line with the analytical SunShell C18 column (150 mm × 100 µm i.d., 90 Å, ChromaNik Technologies Inc., Osaka, Japan), and tryptic (glyco)peptides were separated with 0.1% formic acid as solvent A and 80% acetonitrile (ACN, Honeywell, Charlotte, NC, USA), 0.02% formic acid as solvent B at a flow rate of 1 µL min$^{-1}$ with a linear gradient of 0–80% solvent B in 80 min, in a 90 min analytical run. MS was set to the following parameters: positive ion mode, captive spray with nanoBooster technology, capillary voltage 1 kV, dry gas temperature 150 °C, dry flow 8 L h$^{-1}$. Mass spectra were recorded in the range from m/z 50 to 3000 with a frequency of 0.5 Hz. Three precursors with the highest intensities were automatically selected for CID fragmentation. A search for specific tryptic peptides with a maximum of two miscleavages was done in MaxQuant version 1.6.10.43 software[64] against *Homo sapiens* protein sequences (UniProt fasta file) with the methionine oxidation and asparagine carrying *N*-acetylhexosamine as variable modifications, and

carbamidomethyl on cysteine as the fixed modification. Analysis was performed in triplicates.

**N-glycan release and fluorescent labeling of Tf and IgG glycans**. Dried Tf and IgG eluates after isolation from plasma samples were denatured with 30 µl of 13.3 g L$^{-1}$ sodium dodecyl sulfate (SDS, Invitrogen) and by incubation at 65 °C for 10 min. After cooling down to room temperature for 30 min, 10 µl of 4% (v/v) Igepal CA-630 (Sigma-Aldrich) was added and the mixture was shaken for 15 min on a plate shaker. N-glycans were released after the addition of 10 µL of 5× PBS and 1.2 U of PNGase F (10 U/µL, Promega, Madison, WI, USA) by incubation at 37 °C for 18 h. Released N-glycans were labeled with 2-AB (Sigma-Aldrich). The labeling mixture was freshly prepared by dissolving 2-AB and 2-methylpyridine borane complex (final concentrations 19.2 and 44.8 mg mL$^{-1}$, respectively) in the mixture of dimethyl sulfoxide (Sigma-Aldrich) and glacial acetic acid (Merck) (7:3, v/v). The labeling mixture (25 µL) was added to each sample and the plate was sealed using an adhesive seal. After 10 min of shaking, samples were incubated for 2 h at 65 °C.

**Cleanup of 2-AB-labeled glycans**. Excess reagents from previous steps were removed from the samples using hydrophilic interaction liquid chromatography solid phase extraction (HILIC-SPE). After free N-glycan labeling samples were cooled down to room temperature for 30 min and 700 µL of ACN (previously cooled down to 4 °C) was added to each sample. The cleanup procedure was performed on a hydrophilic 0.2 µm AcroPrep GHP filter plate (Pall Corporation) using a vacuum manifold at around 25 mmHg. All wells of a GHP filter plate were prewashed with 200 µL of 70% (v/v) ethanol in water, 200 µL of ultrapure water, and 200 µL of 96% (v/v) ACN in water (previously cooled down to 4 °C). Diluted samples were loaded to the GHP filter plate wells, and after short incubation subsequently washed with 5 × 200 µL of 96% (v/v) ACN in water. The last washing step was followed by centrifugation at 164×g for 5 min. Glycans were eluted from the plate with 2 × 90 µl of ultrapure water after 15 min shaking at room temperature and centrifugation at 164×g for 5 min in each step. Combined eluates of 2-AB labeled Tf and IgG N-glycans were stored at −20 °C until ultra-high-performance liquid chromatography (UHPLC) analysis.

**Glycan analysis by ultra-high-performance liquid chromatography**. Fluorescently labeled and purified Tf and IgG N-glycans were analyzed by UHPLC based on hydrophilic interactions (HILIC-UHPLC) and detected using excitation and emission wavelengths of 250 and 428 nm, respectively. Acquity UHPLC instrument (Waters) was under the control of Empower 3 software, build 3471 (Waters). Mobile phases were 100 mmol L$^{-1}$ ammonium formate, pH 4.4 (solvent A), and ACN (solvent B) and samples were maintained at 10 °C before injection. Tf 2-AB labeled N-glycans prepared in 75% ACN were separated on a Waters BEH Glycan column, 150 × 2.1 mm i.d., 1.7 µm BEH particles at 25 °C in a 23-min linear gradient of 30–47% solvent A at a flow rate of 0.56 mL min$^{-1}$. IgG 2-AB-labeled N-glycans prepared in 80% ACN were separated on a Waters BEH Glycan column, 100 × 2.1 mm i.d., 1.7 µm BEH particles at 60 °C in a 29-min linear gradient of 25–38% solvent A at a flow rate of 0.4 mL min$^{-1}$.

The HILIC-UHPLC system was calibrated using a dextran ladder (external standard of hydrolyzed and 2-AB labeled glucose oligomers) according to which the retention times for the individual chromatographic peaks (representing the 2-AB-labeled glycan) were converted to glucose units (GU). Data was processed using an automatic processing method with a traditional integration algorithm. Each Tf N-glycans chromatogram integrated into 35 peaks was manually corrected to maintain the same intervals of integration for all the samples, while IgG N-glycans chromatograms were integrated into 24 peaks[20] using automatic integration[65]. The amount of glycans in each chromatographic peak was expressed as a percentage of the total integrated area (% area).

**Exoglycosidase digestions**. A pool of Tf 2-AB-labeled N-glycans (volume equivalent to glycans from 10 µg of Tf) was dried in PCR tubes and processed using the N-Glycan Sequencing Kit according to the manufacturer's instructions (New England Biolabs, Ipswich, MA, USA) (Supplementary Table 3). Each of the eight reactions was done in triplicates. After 18-h digestion at 37 °C reaction mixtures were cleaned up on AcroPrep Omega 96-well filter plate MWCO 10 K NTRL, 350 µL well volume (Pall) using a vacuum manifold at a maximum vacuum. The filter plate was washed 2 × 100 µL of ultrapure water, reaction mixtures (10 µL) were transferred to prewashed wells of the filter plate, and eluates were collected in a clean collection plate. Reaction tubes were rinsed 2 × 20 µL of ultrapure water and transferred in each step to the Omega filter plate. The second and third fractions were collected in the same collection plate resulting in a total volume of 50 µL. Digested Tf N-glycan samples were stored at −20 °C until HILIC-UHPLC analysis, as described.

**Ethyl esterification of Tf 2-AB-labeled N-glycans**. Before mass spectrometry analysis of Tf glycans eluting in each chromatographic peak, the Tf 2-AB labeled N-glycan pool was fractionated by HILIC-UHPLC using the chromatographic method described above. Four equivalent HILIC-UHPLC fractions were combined, dried in a vacuum centrifuge, and reconstituted in 1 µL of ultrapure water. Ethyl

esterification reaction and cotton HILIC glycan enrichment[26] were performed as follows. The volume of 20 μL of 1-ethyl-3-(3-(dimethylamino)propyl)carbodiimide (EDC, AcrosOrganics, Thermo Fisher Scientific)/1-hydroxy-benzotriazole (HOBt, Sigma-Aldrich) reagent in ethanol (0.25 mol L$^{-1}$ each) was added and samples incubated for 1 h at 37 °C on a heated platform without shaking (Heidolph, Schwabach, Germany). ACN (20 μL) was added to each sample, resuspended, incubated at −20 °C for 15 min, and then at room temperature for 15 min. Each cotton pipette tip used for glycan enrichment was prewashed 3× 20 μL of ultrapure water and 3× 20 μL of 85% (v/v) ACN in water. The sample was aspirated and resuspended 20× in the tip. Cotton tip was then washed 3× 20 μL of 85% (v/v) ACN, 1% (v/v) trifluoroacetic acid (TFA, Sigma-Aldrich) and 3× 20 μL 85% (v/v) ACN. The sample was eluted to a clean tube by pipetting 10 μL of ultrapure water and resuspending 20×. Additional elution fraction was collected in the same way, both fractions were combined and dried in a vacuum centrifuge (Martin Christ, Osterode, Germany).

**MALDI-TOF-MS Tf N-glycan analysis**. Chromatographic peak fractions containing 2-AB-labeled ethyl-esterified Tf N-glycans were analyzed by MALDI-TOF-MS. Each sample was reconstituted in 1.5 μL of ultrapure water to which 1.5 μL of matrix solution, containing 5 g L$^{-1}$ 2,5-dihydroxybenzoic acid (2,5-DHB, Bruker Daltonics), 1 mmol L$^{-1}$ sodium hydroxide in 75% (v/v) ACN in ultrapure water, was added. Droplets were mixed by pipetting and 1 μL of the mixture was spotted to MALDI AnchorChip 384 BC (Bruker). Spots were allowed to air dry for 30 min and recrystallized by briefly tapping them with a pipette tip containing 0.2 μL of ethanol.

Analyses were done on an ultrafleXtreme MALDI-TOF-MS (Bruker Daltonics) using positive-ion reflectron mode under the control of Flexcontrol 3.3 software (Bruker Daltonics). Dextran calibration standard (0.8 g L$^{-1}$) was used for instrument calibration before measuring the surrounding eight spotted samples. A mass window of $m/z$ 1000–5000 with suppression up to $m/z$ 900 was used for Tf N-glycans detection. For each spectrum, 3 × 10,000 laser shots were accumulated at a laser frequency of 2000 Hz, using a complete sample random walk with 200 shots per raster spot. Spectra were recorded using 65% laser intensity. Tandem mass spectrometry (MALDI-TOF/TOF-MS/MS) was performed via LIFT$^{TM}$ technology (Bruker Daltonics). Compass DataAnalysis 4.1, build 362.7 (Bruker Daltonics) was used for spectra analysis.

**Tf N-glycan structure assignation**. N-glycan structures were proposed using the GlycoStore database (www.glycostore.org)[29,30] according to UHPLC-HILIC data, GlycoMod Tool (https://web.expasy.org/glycomod/)[66] according to experimentally determined monoisotopic masses (0.5 Da mass tolerance, 143 Da adduct mass that includes 2-AB and Na$^+$ as adducts), results of exoglycosidase array reactions, literature search and established biosynthetic pathways. GlycoWorkbench version 2.1, build 146 was used for MS/MS spectra annotation and depiction of glycan structures on the figures. Figures have been annotated with glycan cartoons following the recommendations of the Consortium for Functional Glycomics[67].

**Statistics and reproducibility**. High-throughput UHPLC data was normalized and batch-corrected to remove experimental variation from glycan measurements. Normalization by total area resulted in % area values expressing the amount of N-glycans in each chromatographic peak. Normalized glycan measurements were then log-transformed due to the right-skewness of their distributions and the multiplicative nature of batch effects. ComBat method (R package sva) was used to perform batch correction on log-transformed measurements, where the technical source of variation (which sample was analyzed on which plate) was modeled as a batch covariate. Estimated batch effects were subtracted from log-transformed measurements to get measurements corrected for experimental noise. In addition to directly measured glycan structures, derived glycosylation traits were calculated based on the shared glycan structural features (e.g. sialylation, galactosylation, fucosylation, etc.) according to Supplementary Tables 4 and 5 for Tf and IgG, respectively.

Method variability was assessed by estimating the measurement error of each Tf chromatographic peak. Estimations are based on the internal standard samples in Korcula ($N = 72$) and VIKING ($N = 71$) cohorts, and based on the sample replicates in Korcula ($N = 2 \times 107 = 214$) cohort. Both internal standards and replicates represent a set of samples that are biologically identical and differences among their measurements are a result of experimental variation. Estimations of the measurement error based on standard samples were calculated as a ratio of the variation of standard samples and variation of cohort samples, multiplied by 100 (100*Var(Stand)/Var(Cohort)). Estimations of the measurement error based on sample replicates were calculated using the formula (1−Correlation(Replicates)) *100.

Associations analyses between glycan traits and biochemical and physiological traits were performed by implementing a general linear regression model or Pearson correlation test (where stated). In analyses where the general regression model was applied, age and sex (and BMI where stated) were included as additional covariates except in those models where age and sex were variables of primary interest. Before regression analyses, glycan variables as well as quantitative biochemical and physiological traits were all transformed to standard Normal

distribution (mean = 0, sd = 1) by an inverse transformation of ranks to Normality (R package "GenABEL", function rntransform). Transformed glycan variables have the same standardized variance so using rank transformed variables in analyses makes estimated effects of different glycans in different cohorts comparable. In analyses where the Pearson correlation test was used, glycan measurements were adjusted for age, sex, and BMI (where stated) prior to correlation analysis. Analyses were firstly performed for each cohort separately ($N$(Korcula) = 927; $N$(VIKING) = 958 for Tf and $N$(Korcula) = 950; $N$(VIKING) = 1087 for IgG) and then combined using fixed-effects meta-analysis approach (R package meta, metagen(method = "FE")). The false discovery rate was controlled using the Benjamini–Hochberg procedure (function p.adjust(method = "BH")). Data were analyzed and visualized using the R programming language (version 4.0.2).

**Reporting summary**. Further information on research design is available in the Nature Portfolio Reporting Summary linked to this article.

## Data availability

There is neither Research Ethics Committee approval, nor consent from individual participants, to permit the open release of the individual-level research data underlying this study. The datasets analyzed during the current study are therefore not publicly available. Instead, the datasets generated and/or analyzed during the current study are available from the corresponding author on reasonable request and in line with the consent given by participants. Supplementary Tables 1–20 can be found in the Supplementary Data 1 file.

## Code availability

The following software packages were used in this study: Empower 3, Flexcontrol 3.3, Compass DataAnalysis 4.1, GlycoWorkbench 2.1, MaxQuant 1.6.10.43, and ComBat. The remaining computer code used to generate results is available from the corresponding author on reasonable request.

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

## Acknowledgements

Viking Health Study—Shetland (VIKING) DNA extractions and genotyping were performed at the Edinburgh Clinical Research Facility, University of Edinburgh. We would

like to acknowledge the invaluable contributions of the research nurses in Shetland, the administrative team in Edinburgh, and the people of Shetland. This work has been supported by funding from the European Structural and Investments Funds for the projects Croatian National Centre of Research Excellence in Personalized Healthcare (contract no. KK.01.1.1.01.0010), Croatian National Centre of Competence in Molecular Diagnostics (contract no. KK.01.2.2.03.0006), and Development of personalized diagnostic tool for prevention and treatment of cardiometabolic disorders—CardioMetabolic (contract no. KK.01.2.1.02.0321). The Viking Health Study—Shetland (VIKING) was supported by the MRC Human Genetics Unit quinquennial program grant "QTL in Health and Disease". The work of L.K. was supported by an RCUK Innovation Fellowship from the National Productivity Investment Fund (MR/R026408/1). For the purpose of open access, the author has applied a Creative Commons Attribution (CC BY) license to any Author Accepted Manuscript version arising from this submission.

## Author contributions

G.L. conceived and supervised the study. I.T.-A., M.P.B., and G.L. coordinated the study. E.R. supervised the MALDI-TOF-MS analyses. A.Š. supervised the development of monolithic plates. I.T.-A., T.P., M.V., U.Č., J.V., J.Š., A.K., and M.N. performed laboratory experiments. F.V. performed data analysis. I.T.-A., F.V., and G.L. wrote the initial draft. L.K., I.K., O.P., and J.F.W. collected and provided samples and data for Korcula and VIKING populations. All authors critically revised and edited the final version of the manuscript.

## Competing interests

The authors declare the following competing interests: G.L. is the founder and owner of Genos Ltd, a private research organization that specializes in high-throughput glycomic analysis and has several patents in this field. I.T.-A., F.V., T.P., M.V., J.Š., M.N., and M.P.B. are employees of Genos Ltd. A.Š. is the founder and managing director of BIA Separations Ltd, a Sartorius company, focused on the development of monolith technology and production of CIM® (convective interaction media) chromatographic columns for the production, purification, and analysis of large biomolecules. U.Č. and J.V. are employees of BIA Separations Ltd. E.R. is co-affiliated with glyXera GmbH. glyXera provides high-performance glycoanalytical products and services and holds several patents for xCGE-LIF and MALDI-MS-based glycoanalysis. The other authors declare no competing interests.
