## [Peer Review File · Communications Biology]

Reviewers' comments:

Reviewer #1 (Remarks to the Author):

Purpose of the research, originality, and significance:

This manuscript does an excellent and comprehensive investigation into the glycosylation of Transferrin (Tf). The manuscript is the first in-depth analysis of the glycosylation of human serum transferrin. Glycosylation is known to change due to other criteria in addition to disease. Thus, the result of this research is important to those investigating the glycosylation of transferrin as a potential biomarker. In addition, it will add to the current body of literature regarding the dynamic nature of protein glycosylation in health and disease. The results of this study will be of interest to the glycomics and glycoproteomic scientific community.

Major claims/findings of the paper:

The authors address issues surrounding the purification of Tf and propose a high-throughput method for isolation, thus allowing for the first large-scale study of Tf. The purification process uses standard protein purification techniques consisting of highly abundant glycoproteins prior to a more specific antibody solid phase extraction. The purification process resulted in a sample with 99.36% purity.

- Protein purity is essential to this study; the purity was determined through MS analysis and is convincing.

Tf glycosylation varied with age, showing an increase in structural complexity regardless of gender (<10%). What is of interest is that the difference due to age was substantially less than what is seen for IgG. IgG glycosylation due to age is well documented.

- The findings are of interest since, outside of IgG, there are few serum proteins that have been studied regarding the effect of aging on protein-specific glycosylation.

The opposite was observed with sex where Tf showed a greater difference (10%) between the male and female sexes when compared to IgG (5%).

- The researchers should note that the Tf sex difference, when compared to Tf age-related changes, is similar (10% or less).

Tf N-glycomes variance occurred due to several traits attributed to an unhealthy lifestyle. The associations were stronger for Tf than for IgG except for smoking and menstruation.

- These findings are of interest and add to the body of literature as researchers work toward understanding the cause/purpose/function of aberrant glycosylation patterns in otherwise very steady protein-specific glycan profiles found in healthy individuals.

The authors address the issue of many traits having strong associations with age and sex and BMI and thus the analysis was performed with these as covariates. Many medical conditions were not found to be statistically significant contributors to the variability while most biochemical parameters remained significant.

- Because of the comorbidities associated with obesity, this was an important distinction to make and adds to the credibility of the findings.

Conclusions:

The authors hypothesize that differences found in aberrant IgG and Tf glycosylation could reflect functionality. While the Tf glycosylation link to function is vigorous and well-cited, the IgG discussion would benefit from the following suggestions.

- The authors do not discuss the role of IgG glycosylation on pro- and anti-inflammatory immune responses or receptor binding, but only mention it reflects previous antigen encounters during past infections. The authors should (briefly) discuss the known effects of IgG glycosylation on the pro- and anti-inflammatory effects of the molecule. They may also give thought to the decreased immune function of the elderly. Do the authors have evidence for IgG glycosylation profiles as an indication of past infections? If so, please cite. If this is a hypothesis put forth by the authors, it should be supplemented by a rationale.

Data & methodology:

A major strength of this manuscript is the large sample size and the analysis of two populations. The authors relied on proven glycan identification and quantification techniques, including 2AB-labeled UPLC glycan profiles (GU values), exoglycosidase sequencing, and MALDI-MS.

- The researchers provide detailed methods and a description of the statistical analysis that allows the study to be reproduced. Furthermore, the statistical analysis was suitable and robust. The supplementary data is comprehensive and complete.
- The authors should clearly identify the internal standard/source used for estimated measurement.

Reviewer #2 (Remarks to the Author):

In the manuscript entitled "COMPARATIVE ANALYSIS OF TRANSFERRIN AND IgG N-GLYCOSYLATION IN TWO HUMAN POPULATIONS," the authors developed a high throughput platform using a 96-well monolithic plate with antibodies to purify transferrin. In addition, the developed method was applied to large-scale samples, which provide solid information about glycosylation profiles.

The developed method prevents contamination from IgG, which is more selective compared to other methods. The application results are informative, which benefits our understanding of transferrin glycosylation. Only one suggestion regarding the performance of the 96-well monolithic plate. Can the authors provide information such as purification repeatability and purification capacity?

Reviewer #3 (Remarks to the Author):

Trbojević-Akmačić and colleagues present a manuscript describing a large-scale comparative study in healthy populations to assess possible associations between Transferrin and Immunoglobulin G N-glycome levels and multiple biochemical and physiological traits. Results show a strong correlation between Tf glycome and age and gender of the subjects involved in the study.

The manuscript is well written and interesting to read, the analytical techniques for sample characterization are robust and reliable, high-throughput IgG and Tf isolation protocols, as well as analytical methods, are described in detail, sample size and statistical analysis are appropriate. The manuscript is ready for publication without the need of major revision.

Minor issue

Since this study analyzes human samples, perhaps it is better to use the term "gender" instead of "sex".

RESPONSE TO THE REVIEWERS' COMMENTS

REVIEWER #1

COMMENTS:

Purpose of the research, originality, and significance:

This manuscript does an excellent and comprehensive investigation into the glycosylation of Transferrin (Tf). The manuscript is the first in-depth analysis of the glycosylation of human serum transferrin. Glycosylation is known to change due to other criteria in addition to disease. Thus, the result of this research is important to those investigating the glycosylation of transferrin as a potential biomarker. In addition, it will add to the current body of literature regarding the dynamic nature of protein glycosylation in health and disease. The results of this study will be of interest to the glycomics and glycoproteomic scientific community.

Major claims/findings of the paper:

The authors address issues surrounding the purification of Tf and propose a high-throughput method for isolation, thus allowing for the first large-scale study of Tf. The purification process uses standard protein purification techniques consisting of highly abundant glycoproteins prior to a more specific antibody solid phase extraction. The purification process resulted in a sample with 99.36% purity.

- Protein purity is essential to this study; the purity was determined through MS analysis and is convincing.

Tf glycosylation varied with age, showing an increase in structural complexity regardless of gender (<10%). What is of interest is that the difference due to age was substantially less than what is seen for IgG. IgG glycosylation due to age is well documented.

- The findings are of interest since, outside of IgG, there are few serum proteins that have been studied regarding the effect of aging on protein-specific glycosylation.

The opposite was observed with sex where Tf showed a greater difference (10%) between the male and female sexes when compared to IgG (5%).

- The researchers should note that the Tf sex difference, when compared to Tf age-related changes, is similar (10% or less).

Tf N-glycomes variance occurred due to several traits attributed to an unhealthy lifestyle. The associations were stronger for Tf than for IgG except for smoking and menstruation.

- These findings are of interest and add to the body of literature as researchers work toward understanding the cause/purpose/function of aberrant glycosylation patterns in otherwise very steady protein-specific glycan profiles found in healthy individuals.

The authors address the issue of many traits having strong associations with age and sex and BMI and thus the analysis was performed with these as covariates. Many medical conditions were not found

to be statistically significant contributors to the variability while most biochemical parameters remained significant.

- Because of the comorbidities associated with obesity, this was an important distinction to make and adds to the credibility of the findings.

Conclusions:

The authors hypothesize that differences found in aberrant IgG and Tf glycosylation could reflect functionality. While the Tf glycosylation link to function is vigorous and well-cited, the IgG discussion would benefit from the following suggestions.

- The authors do not discuss the role of IgG glycosylation on pro-and anti-inflammatory immune responses or receptor binding, but only mention it reflects previous antigen encounters during past infections. The authors should (briefly) discuss the known effects of IgG glycosylation on the pro- and anti-inflammatory effects of the molecule. They may also give thought to the decreased immune function of the elderly. Do the authors have evidence for IgG glycosylation profiles as an indication of past infections? If so, please cite. If this is a hypothesis put forth by the authors, it should be supplemented by a rationale.

Author reply: We thank the Reviewer for the time invested in carefully checking our manuscript and for suggestions to improve our manuscript. The Discussion section was expanded with a paragraph on the role of IgG glycosylation in inflammation (lines 331-344), as follows:

“The role of IgG N-glycans in the modulation of its effector functions has been well recognized⁴⁹⁻⁵². One of the most studied examples is the addition of core fucose to the Fc fragment Asn297 N-glycosylation site that decreases antibody-dependent cell-mediated cytotoxicity (ADCC) up to 100-fold through lower IgG affinity toward Fcγ receptor IIIa^{53,54}. Moreover, both galactosylation and sialylation have been shown to modulate the inflammatory activity of IgG. While galactosylated IgG has an anti-inflammatory effect by inhibiting the complement pathway⁵⁵, agalactosylated IgG glycans act pro-inflammatory by activating the alternative pathway⁵⁶, as well as lectin complement pathway via interaction with mannose-binding lectin⁵⁷. Sialylation also changes IgG activity from pro-inflammatory to anti-inflammatory^{31,51}, although the exact mechanisms remain to be elucidated. Pro-inflammatory glycosylation profile of IgG, predominantly characterized by decreased galactosylation and sialylation, has been seen with aging^{21,58}, as well as with different diseases²³. It was hypothesized that the lifelong antigenic challenges and increasing antigenic burden cause immune system remodeling leading to the state of a low-grade chronic inflammation that characterizes aging, i.e. inflammaging⁴⁶. In this context, pro-inflammatory IgG glycans are considered both biomarkers and functional effectors of aging⁴⁷.”

Moreover, the sentence “...so it is conceivable IgG glycosylation correlates more with age, than with sex, since it somewhat reflects previous antigen encounters during past infections.” was rephrased to “...so it is conceivable IgG glycosylation correlates more with age, than with sex, since it was proposed that it somewhat reflects previous antigen encounters during past infections⁴⁶⁻⁴⁸.” with appropriate references (lines 328-330).

Data & methodology:

A major strength of this manuscript is the large sample size and the analysis of two populations. The authors relied on proven glycan identification and quantification techniques, including 2AB-labeled UPLC glycan profiles (GU values), exoglycosidase sequencing, and MALDI-MS.

- The researchers provide detailed methods and a description of the statistical analysis that allows the study to be reproduced. Furthermore, the statistical analysis was suitable and robust. The supplementary data is comprehensive and complete.
- The authors should clearly identify the internal standard/source used for estimated measurement.

Author reply: We thank the Reviewer for the comment. Sources of internal plasma standard samples used for quality control and batch correction were mentioned in the Methods – Experimental design section, as suggested (lines 390-393):

“Standard plasma sample used for Korcula analysis was prepared as a pool of cohort samples. Standard plasma sample used for VIKING cohort analysis was obtained from the Croatian National Institute of Transfusion Medicine after approval by the Ethical Committee of the institute.”

REVIEWER #2

COMMENTS:

In the manuscript entitled "COMPARATIVE ANALYSIS OF TRANSFERRIN AND IgG N-GLYCOSYLATION IN TWO HUMAN POPULATIONS," the authors developed a high throughput platform using a 96-well monolithic plate with antibodies to purify transferrin. In addition, the developed method was applied to large-scale samples, which provide solid information about glycosylation profiles.

The developed method prevents contamination from IgG, which is more selective compared to other methods. The application results are informative, which benefits our understanding of transferrin glycosylation. Only one suggestion regarding the performance of the 96-well monolithic plate. Can the authors provide information such as purification repeatability and purification capacity?

Author reply: We thank the Reviewer for their valuable time and suggestions to improve our manuscript. Information about purification repeatability and purification capacity of the CIMac-@Tf 96-well plate was added in the Results section, as suggested (lines 104-107):

“Average purification capacity of the CIMac-@Tf 96-well plate was 300 µg of Tf per well (coefficient of variation, CV = 9.1% for the plate) ²⁸. Tf purification repeatability within the cohorts was determined from the standard plasma samples and resulted in a CV of 18.9% (average Tf mass = 156 ± 30 µg, n = 72) for Korcula and 23.4% (average Tf mass = 85 ± 20 µg, n = 79) for VIKING.”

REVIEWER #3

COMMENTS:

Trbojević-Akmačić and colleagues present a manuscript describing a large-scale comparative study in healthy populations to assess possible associations between Transferrin and Immunoglobulin G N-glycome levels and multiple biochemical and physiological traits.

Results show a strong correlation between Tf glycome and age and gender of the subjects involved in the study.

The manuscript is well written and interesting to read, the analytical techniques for sample characterization are robust and reliable, high-throughput IgG and Tf isolation protocols, as well as analytical methods, are described in detail, sample size and statistical analysis are appropriate. The manuscript is ready for publication without the need of major revision.

Minor issue

Since this study analyzes human samples, perhaps it is better to use the term "gender" instead of "sex".

Author reply: We thank the Reviewer for their valuable time and suggestion to improve our manuscript. Regarding the term "sex" used in the manuscript, we consider it to refer to the different biological and physiological characteristics of males and females and not to a person's self-representation influenced by social, cultural, and personal experience, for which we believe "gender" would be a more appropriate term. Therefore, the term "sex" was used throughout the manuscript.